# Landscape of Bone Marrow Metastasis in Human Neuroblastoma Unraveled by Transcriptomics and Deep Multiplex Imaging

**DOI:** 10.3390/cancers13174311

**Published:** 2021-08-26

**Authors:** Daria Lazic, Florian Kromp, Fikret Rifatbegovic, Peter Repiscak, Michael Kirr, Filip Mivalt, Florian Halbritter, Marie Bernkopf, Andrea Bileck, Marek Ussowicz, Inge M. Ambros, Peter F. Ambros, Christopher Gerner, Ruth Ladenstein, Christian Ostalecki, Sabine Taschner-Mandl

**Affiliations:** 1St. Anna Children’s Cancer Research Institute (CCRI), 1090 Vienna, Austria; daria.lazic@ccri.at (D.L.); florian.kromp@scch.at (F.K.); fikret.rifatbegovic@ccri.at (F.R.); peter.repiscak@ccri.at (P.R.); Mivalt.Filip@mayo.edu (F.M.); florian.halbritter@ccri.at (F.H.); marie.bernkopf@ccri.at (M.B.); inge.ambros@ccri.at (I.M.A.); peter.ambros@ccri.at (P.F.A.); ruth.ladenstein@ccri.at (R.L.); 2Software Competence Center Hagenberg (SCCH), 4232 Hagenberg, Austria; 3Department of Dermatology, University Hospital Erlangen, 91054 Erlangen, Germany; kirr@students.uni-marburg.de (M.K.); Christian.Ostalecki@uk-erlangen.de (C.O.); 4Department of Analytical Chemistry, Faculty of Chemistry, University of Vienna, 1090 Vienna, Austria; andrea.bileck@univie.ac.at (A.B.); christopher.gerner@univie.ac.at (C.G.); 5Department and Clinic of Pediatric Oncology, Hematology and Bone Marrow, Transplantation, Wroclaw Medical University, 50-556 Wroclaw, Poland; ussowicz@o2.pl

**Keywords:** bone marrow, metastasis, neuroblastoma, tumor heterogeneity, tumor microenvironment, single-cell analysis, multiplex imaging, deep learning, transcriptomics, disseminated tumor cells

## Abstract

**Simple Summary:**

Bone marrow metastasis frequently occurs in patients with solid cancers and most often leads to poor outcome. Yet, the composition of bone marrow metastases, including tumor and surrounding cells, has so far not been characterized. Herein, we aimed to investigate the diversity of tumor and surrounding cells, i.e., the microenvironment, in bone marrow metastases, using the childhood tumor neuroblastoma as a model. To this end, we screened genome-wide datasets to define a panel of cell-specific markers for multiplex microscopy of metastatic bone marrow samples, and developed DeepFLEX, a computational pipeline for subsequent image analysis. Thereby, we identified 35,000 single cells covering metastasized tumor cells, and various types of developing immune and bone marrow cells. In parallel, we analyzed the transcriptome, i.e., all genes that are expressed as mRNA, of 38 patients with and without bone marrow metastasis. We found vast tumor cell diversity and identified a marker protein, FAIM2, which can help to identify a broader range of tumor cell variants. In addition we showed that tumor cell metastasis in the bone marrow is associated with an immune response resembling inflammation, and the presence of cells that can repress an immune attack against cancer cells. Our study suggests that metastatic tumor cells are shaping the bone marrow microenvironment and builds the basis to further investigate its clinical relevance.

**Abstract:**

While the bone marrow attracts tumor cells in many solid cancers leading to poor outcome in affected patients, comprehensive analyses of bone marrow metastases have not been performed on a single-cell level. We here set out to capture tumor heterogeneity and unravel microenvironmental changes in neuroblastoma, a solid cancer with bone marrow involvement. To this end, we employed a multi-omics data mining approach to define a multiplex imaging panel and developed DeepFLEX, a pipeline for subsequent multiplex image analysis, whereby we constructed a single-cell atlas of over 35,000 disseminated tumor cells (DTCs) and cells of their microenvironment in the metastatic bone marrow niche. Further, we independently profiled the transcriptome of a cohort of 38 patients with and without bone marrow metastasis. Our results revealed vast diversity among DTCs and suggest that FAIM2 can act as a complementary marker to capture DTC heterogeneity. Importantly, we demonstrate that malignant bone marrow infiltration is associated with an inflammatory response and at the same time the presence of immuno-suppressive cell types, most prominently an immature neutrophil/granulocytic myeloid-derived suppressor-like cell type. The presented findings indicate that metastatic tumor cells shape the bone marrow microenvironment, warranting deeper investigations of spatio-temporal dynamics at the single-cell level and their clinical relevance.

## 1. Introduction

Metastasis is the major cause of cancer-related deaths [1] and relies on the ability of tumor cells to disseminate from the primary site and adapt to distant tissue environments [2]. This is an arduous process, which can fuel heterogeneity among metastasizing and disseminated tumor cells (DTCs) [3]. Tumor heterogeneity manifests in variations of clinically important features such as the abundance of prognostic markers as well as therapeutic targets, which complicates patient stratification and explains failure of therapeutic approaches [4,5,6,7].

Cancer cells are attracted by distant microenvironments that promote their growth and survival [8]. One such hospitable microenvironment is the bone marrow, which has a major role in dormancy and relapse [9] and is a frequent site of dissemination in numerous solid cancers [10,11], such as breast cancer, colorectal cancer and neuroblastoma [12].

Neuroblastoma, an extracranial neoplasm of the sympathetic nervous system, is the most common solid tumor in patients in their first year of life and accounts for roughly 15% of childhood cancer related deaths [13,14,15]. In more than 90% of metastatic stage (stage M) neuroblastoma patients, tumor cells disseminate to the bone marrow [16,17], where some tumor cells may resist initial chemotherapy and give rise to relapse. These relapse seeding clones are frequently detected in the bone marrow already at the time-point of diagnosis, but not in the primary tumor [4]. Based on bulk RNA-sequencing (RNA-seq), we have previously shown differences between the transcriptome of DTCs with predominantly hypoxia-associated genes enriched, and primary tumor cells with an increased expression of mesenchymal genes [18]. Subsequently, two studies of neuroblastoma cell lines and primary tumors unraveled the gene regulatory networks driving two plastic phenotypes, adrenergic and mesenchymal type neuroblastoma cells, and highlighted their importance, as the latter were more frequently found in post-therapy and relapse samples and were more resistant to chemotherapy [19,20]. Thus, genetic and phenotypic tumor heterogeneity can be considered key to why treatment outcome of metastatic disease remains poor.

Phenotypic heterogeneity of solid cancers has been investigated at the primary site at single-cell resolution [21,22,23,24] and recently single cell RNA-sequencing has been employed to identify the cell of origin and developmental trajectories in primary neuroblastoma tumors [25,26,27,28,29]. To date, only a few reports have studied bone marrow metastases in animal models [30], but no analyses in humans have been undertaken. 

In recent years, numerous technologies for the analysis of single cells have emerged and advanced rapidly. While single-cell RNA-seq (scRNA-seq) methods [31] enable high-dimensional analyses of cells at the transcriptomic level, highly multiplexed imaging methods [32] provide an image of every cell and thereby allow subcellular localization of proteins as well as morphological assessment. Despite the volume of developing multiplex imaging methods, the standard method to detect DTCs in bone marrow aspirates in neuroblastoma routine diagnostic procedures, is still automated immunofluorescence plus fluorescence in situ hybridization (AIPF), which is limited to only three biomarkers (GD2, CD56 and one genetic marker) [33,34]. In order to gain deeper insights into intra-tumor heterogeneity and capture metastasis-related changes in the bone marrow environmental composition, a comprehensive single-cell map of bone marrow metastases is indispensable. Thus, we sought to provide the first single-cell atlas of bone marrow metastases including DTCs and cells of the microenvironment, by employing neuroblastoma as a model.

Based on multi-omics data mining, we established a 20-plex antibody panel and applied Multi-Epitope-Ligand Cartography (MELC), a multiplex imaging method with a resolution of 450 nm that employs automated sequential cycles of staining with fluorophore-coupled antibodies followed by immunofluorescence (IF) microscopy and photobleaching [35,36]. We developed and validated an image analysis pipeline, called DeepFLEX, which tackles frequent obstacles of IF-based imaging and in addition involves accurate, deep learning-based cell and nucleus segmentation. In parallel, we separated the cells of the bone marrow microenvironment from DTCs and profiled the transcriptome of 38 independent tumor-cell infiltrated and non-infiltrated samples. Our study delivered the first indication that metastasis of neuroblastoma tumor cells into the bone marrow is associated with large microenvironmental adaptations, i.e., pro-inflammatory activation, but at the same time upregulation of regulatory/suppressive signatures and cell types, most importantly an immature neutrophil, granulocytic myeloid-derived suppressor-like cell type. Moreover, we revealed novel markers, including FAIM2 (Fas Apoptotic Inhibitory Molecule 2), to better capture heterogeneity of DTCs in bone marrow metastases of neuroblastoma patients.

## 2. Methods

### 2.1. Subject Details

All subjects of this study are described in detail in Section 2.1.1, Section 2.1.2 and Section 2.1.3.

#### 2.1.1. Neuroblastoma Cell Lines

Five patient-derived neuroblastoma cell lines were used for validation of biomarkers and identification of the best sample preparation conditions. STA-NB-2, -4 and -10 were established from primary tumors, and STA-NB-8 and STA-NB-12 from DTCs of bone marrow aspirates. INSS (International Neuroblastoma Staging System) and *MYCN* amplification status for all neuroblastoma cell lines were described previously [37] and are listed in Appendix A. Cells were maintained in RPMI1640-Glutamax-I (GIBCO) supplemented with 1% Pen/Strep (GIBCO), 10% FCS (PAA Laboratories), 1 mM sodium pyruvate (PAN Biotech) and 25 mM HEPES (PAN Biotech). All neuroblastoma cell lines were cultivated at 37 °C and 5% CO_2_.

#### 2.1.2. Bone Marrow Aspirates

Bilateral bone marrow aspirates were collected according to the SIOPEN/HR-NBL-1 study protocol or standard of care during routine diagnostics at initial diagnosis and at clinical response evaluation time points. Samples were shipped at room temperature within 4 h or at 4 °C within 24 h. Bone marrow-derived mononuclear cells (MNCs) were isolated by density gradient centrifugation (LymphoprepTM, AXIS-SHIELD PoC AS).

For the validation of antibodies, neuroblastoma cell lines were mixed with tumor-free bone marrow-derived MNCs to obtain a tumor cell suspension of 5% neuroblastoma cell line in bone marrow-derived MNCs.

For single-cell analysis, eight bone marrow aspirates (Appendix A) were collected at different time points along the therapy protocol from four neuroblastoma patients with metastatic (INRG stage M or Ms) disease. For bulk RNA-sequencing for this study 38 bone marrow aspirates from 38 patients with (*n* = 17) and without (*n* = 21) bone marrow infiltration were processed as previously described [18] (Appendix A). Annotations of common neuroblastoma genetic aberrations for the four patients are given in Appendix A.

#### 2.1.3. Peripheral Blood-Derived MNCs

Leftover samples of peripheral blood from routine diagnostics were collected and peripheral blood-derived MNCs were isolated, washed and counted as described for bone marrow-derived MNCs. Neuroblastoma cell lines were spiked into peripheral blood-derived MNCs to obtain a tumor cell content of 5%. The cell mixture was cultivated in the presence of 0.125% (*v*/*v*) anti-CD3/CD28 beads (Thermo Fisher Scientific, Waltham, MA, USA) and 1% (*v*/*v*) IFNγ (Peprotech) for 5 days.

### 2.2. Biomarker Identification by Data Mining

DTC-associated biomarkers were identified based on data mining of previously generated RNA-seq datasets and proteomics data, and guided by public databases. The employed prioritization scheme is detailed in the supplementary methods (Appendix A).

### 2.3. Biomarker Validation

Methods applied to validate selected biomarkers are detailed in the supplementary methods.

### 2.4. Multi Epitope Ligand Cartography

MELC was employed for multiplex IF-staining of the herein established 20-plex antibody panel, as described [36].

Briefly, MELC is based on repetitive cycles of antibody staining and photobleaching. After system start, four fields of view are selected and calibration (brightfield and darkframe) images are acquired. Prior to every staining and photobleaching cycle with the acquisition of the corresponding fluorescence tag and post-bleaching image, the slide is washed with PBS and a phase-contrast image is taken.

Camera (ApogeeKX4, Apogee Instruments) and light source maintain the same position; the motor-controlled xy stage of the inverted fluorescence microscope (Leica DMIRE2, Leica Microsystems; x20 air lens; numerical aperture, 0.7) moves in between fields of view. Images with a resolution of 2018 × 2018 pixels are acquired, with one pixel corresponding to 0.45 µm at a 20× magnification. Thus the whole image covers a field of view covering 908.1 × 908.1 µm.

Additionally, negative control secondary antibodies were implemented, which were applied to the sample prior to indirect staining of the respective primary antibody.

The subsequently applied interphase fluorescence in situ hybridization (iFISH) is described in the supplementary methods.

### 2.5. Sample Preparation, RNA-Isolation and RNA-Sequencing

The dimethyl sulfoxide (DMSO) frozen MNC samples of bone marrow aspirates were stored in liquid nitrogen. Upon thawing, density gradient centrifugation (LymphoprepTM, Axis-Shield, Dundee, UK) was performed in order to remove dead cells. Following the density gradient separation, the GD2 positive DTC fraction was separated from the GD2 negative MNC fraction at 4 °C as described earlier [38]. In brief, the MNC fraction containing the GD2 positive DTC fraction was washed with PBS at 300 g for 10 min at 4 °C. The supernatant was discarded and the cells were re-suspended in 2 mL ice-cold magnetic-activated cell sorting (MACS) buffer (PBS pH 7.2, 0.5% bovine serum albumin, 2 mM ethylene diamine tetraacetic acid). Afterwards, the cells were incubated with 2.5 µL FITC-labeled anti-GD2 antibodies (14.18 delta CH2 clone) at 4 °C for 20 min, followed by a 15 min incubation with 75 µL anti-FITC magnetic beads (Miltenyi, Bergisch Gladbach, Germany). Finally, the DTC-depleted MNC fraction was collected during the MACS sorting at 4 °C and immediately homogenized in QIAzol (Qiagen, fortuna dusseldorf, Germany). The total RNA was isolated with the miRNeasy Micro Kit (Qiagen) following the manufacturer’s protocol. The quantity and integrity were assessed by the Qubit RNA HS Assay Kit (Life Technologies, Foster, CA, USA) and the TapeStation High Sensitivity RNA ScreenTape (Agilent, Santa Clara, CA, USA). For RNA-Seq, 30 ng of total RNA was used for cDNA synthesis following the NEBNext Ultra RNA (non-directional protocol) and NEBNext Ultra RNA II (directional protocol) library preparation kits (New England BioLabs, Ipswich, MA, USA). Upon cDNA synthesis, the library was completed in an automated way at the EMBL Genomics Core Facility (Heidelberg, Germany). RNA-Seq was performed at the Illumina HiSeq 2000 platform by pooling six samples per lane and generating 50 bp single-end reads.

### 2.6. DeepFLEX

The main parameters used by methods integrated into the pipeline are listed below. Further parameters are detailed in Appendix A.

#### 2.6.1. Image Processing

Images were registered, as previously described [39] (Appendix A). Then, flat field correction using brightfield and dark frame calibration images was performed to eradicate gross variations in illumination (Appendix A). Accumulative background noise caused by residual post-bleaching signals was eliminated by subtracting post-bleaching images from successive fluorescence tag images (Appendix A). To reduce vignetting (reduction in image brightness toward periphery compared to image center), intensity distributions were corrected using regularized energy minimization on the set of all fluorescence tag images from our eight samples via CIDRE v0.1 [40] (Appendix A).

#### 2.6.2. Segmentation

For accurate nuclei and cell segmentation, annotated datasets [41] of propidium iodide or phase-contrast images were created, respectively. We trained the deep learning architecture Mask R-CNN for instance-aware segmentation, as previously described [42]. Briefly, after augmenting the training dataset with automatically generated artificial images, we used image tiling and rescaling to segment MELC images in order to make them compatible with the input size (256 × 256 pixels) of the trained Mask R-CNN. The phase contrast image of the propidium iodide staining cycle (which is acquired prior to each IF staining) was segmented into a labeled cell mask (Appendix A), while the fluorescence tag image (nuclear stain propidium iodide) was segmented into a labeled nucleus mask (Appendix A). Inferred objects were only counted as cells if they were reproduced in the nucleus mask (Appendix A). We furthermore removed cells affected by image artifacts or located in poorly illuminated image corners by user-guided region selection (Appendix A).

#### 2.6.3. Feature Extraction

The segmentation masks were used as a reference to generate multi-channel single-cell images (Appendix A), based on which intensity and morphological features were extracted (Appendix A, Appendix A). The morphology of the cell nucleus was described by the features size, perimeter, roundness and solidity. To describe the morphology of the cell, the size and perimeter of the features were extracted. We used three intensity features to quantitate marker abundance: mean intensity, total intensity and mean of the top 20% intensities (less dependent on cell size). Intensity was measured per cell, per nucleus, and per cell cytoplasm and membrane (= cell − nucleus).

#### 2.6.4. Normalization

To eliminate unspecific staining caused by secondary antibodies and to increase the signal-to-noise ratio, features extracted from the second secondary antibody were divided by features extracted from the negative control secondary antibody (Appendix A).

To further remove the unspecific binding of primary antibodies and batch variation in staining intensity, autofluorescence, and illumination, we applied RESTORE v0.1 [43] (Appendix A) to predict a background level (threshold separating signal and noise/background) for each marker in each image based on a mutually exclusive counterpart (Appendix A). The application of RESTORE is detailed in the supplementary methods.

#### 2.6.5. Feature Validation

Clustering stability and specificity were evaluated based on the first field of view from patient sample BM 1.1 containing 2021 cells. The evaluation was performed using consensus clustering on single-cell feature vectors comprised of (I) all three intensity features in the cell, nucleus and cytoplasm/membrane with the morphological features, (II) the mean of the 20% highest pixel intensities in all three cell compartments with morphological features, (III) the mean intensity in all three cell compartments with morphological features, (IV) the total intensity in all three cell compartments with morphological features and (V) the mean of the 20% highest pixel intensities in all three cell compartments without morphological features. Consensus clustering was carried out with the R/Bioconductor package “cola” v1.6.0 [44] repeating Gaussian mixture modelling for model-based clustering (partition method = mclust) of t-distributed stochastic neighbor embedded (perplexity = 30) and z-score standardized data into ten clusters (G = 10) 100 times, each time randomly subsampling the feature space with a sampling rate of 0.8. Clustering stability and specificity were measured by the three provided metrics proportion of ambiguous clustering (PAC), concordance (CON) and mean silhouette score (MSS) and their mean (mean clustering score, MCS) defined as:

(1)
$$\mathrm{MCS}=\frac{\left(1-\mathrm{PAC}\right)+\mathrm{CON}+\mathrm{MSS}}{3}$$


Significantly different feature values between clusters were determined using the F-test, as described by Gu et al. Principal component analysis (PCA) with the R package FactoMineR v2.4 and FactoExtra v1.0.7 [45] was carried out on the autoscaled single-cell feature vectors comprised of the mean of the 20% highest pixel intensities of all 20 markers with the morphological features from all ten determined clusters combined, and on the autoscaled single-cell feature vectors comprised of the mean of the 20% highest pixel intensities of the DTC markers (CD276, GD2, CD24, CD56, FAIM2) with the morphological features from the five DTC clusters (cluster 1, 4, 5, 7 and 9) only.

#### 2.6.6. Single-Cell Analysis

Normalized features were converted into an FCS file format and loaded into cytosplore v.2.3.1 [46]. A-tSNE (approximated and user steerable t-distributed Stochastic Neighbor Embedding, *perplexity* = 30) [47] and subsequent clustering by GMS (Gaussian Mean Shift, *σ* = 45) [48] clustering was computed on the complete single-cell dataset of eight bone marrow samples and resulted in 10 clusters, which were exported as FCS files together with the CSV file of the corresponding heatmap. The latter were imported into python v3.7 to allow further quantitative and explorative analysis with the python data visualization library seaborn v0.10.rc0 [49] (Appendix A). Hierarchical clustering of DTCs into 30 sub-clusters was performed using the complete-link/Voorhees algorithm [48] provided by seaborn v0.10.

### 2.7. RNA-Sequencing Analysis

Quality checks and trimming of the raw RNA-Seq data files were conducted using FastQC v0.11.9 [50] and Trim Galore v0.6.6 [51]. RNA-Seq single-end reads were aligned to the Ensembl GRCh38.p13 v102 [52] human genome using HiSat2 v2.2.1 [53] and reads were assigned to gene counts using featureCounts v2.0.1 [54]. Data analysis was performed in R v4.0.3 [55] using a combination of tidyverse [56], stats and variancePartition v1.20.0 [57] packages for data manipulation and exploratory data analysis. This was followed by differential expression analysis (Appendix A) using DESeq2 v1.30.0 [58] package and Gene Set Enrichment Analysis (GSEA) carried out using clusterProfiler v3.18.1 [59] with fgsea v1.16.0 [60] algorithm. The GSEA genes from the differential expression analysis were ranked according to the log2 fold change and converted to entrez_gene_ids using biomaRt [61] package. The GSEA enrichment was done against different geneset collections obtained from the Molecular Signatures Database (MSigDB) [62,63,64] using msigdbr v7.2.1 [65] package (Appendix A). Tumor microenvironment cell estimation was performed with gsva function from the GSVA v1.38.2 [66] package specifying ssgsea method [67] and using the HAY_BONE_MARROW [68] genesets from the MSigDB.

After detailed exploratory data analysis, we identified a strong batch effect (denoted in the text as library_type; U—unstranded and SR—stranded protocol) originating from different RNA-seq preparation protocols and preparation batches. In order to account for the batch effect in the downstream analyses, it was either included in the DESeq2 model for differential expression analysis or, where batch corrected counts were required, it was removed using the removeBatchEffect() function from the limma [69] package. To further ameliorate possible influences of batch effects, we repeated the analysis with/without batch correction and focused on results that were consistent between both analyses (data not shown).

## 3. Results

### 3.1. Comprehensive Single-Cell Multiplex Immunofluorescence Imaging Panel

To analyze bone marrow metastases on a single-cell level, we sought to establish a MELC panel specific to neuroblastoma DTCs and the bone marrow microenvironment, which is mostly composed of hematopoietic and mesenchymal (stromal) cells.

Therefore, in our workflow (Figure 1a), we first selected DTC-associated biomarkers in a multi-omics data mining approach, which we then validated separately by conventional IF staining. Data mining based on RNA-seq data of stage M neuroblastoma primary tumors, DTCs, and bone marrow-derived mononuclear cells (MNCs); proteomics data of neuroblastoma tumors, neuroblastoma cell lines, and peripheral-nerve-associated fibroblasts; and public databases (Uniprot, Protein Atlas, PubMed) revealed 12 DTC-related biomarkers that were highly or exclusively expressed by DTCs and primary tumor cells and localized to the cytoplasmic membrane (Figure 1b,c) [20,70,71,72,73]. These 12 DTC-related biomarkers were tested using optimized IF-sample preparation protocols in a systematic validation procedure (Appendix A) on neuroblastoma cell lines in isolation (Appendix A) and spiked into MNCs (Figure 1d and Appendix A). Upon validation (Appendix A), FAIM2, which was highly abundant by proteomics specifically in neuroblastoma cells (Appendix A), GD2, CD56, VIM and B7-H3 were selected for multiplex imaging of DTCs (Figure 1e).

Second, in order to explore potential changes in the bone marrow microenvironment, we selected 14 protein markers for cell types found in the adult bone marrow, i.e., bone marrow hematopoietic stem and progenitor cells, T-helper (Th) cells, cytotoxic T cells (CTL), regulatory T-cells (Treg), B-cells, monocytes/macrophages, granulocytes, natural killer (NK)-cells and mesenchymal cells [11,74,75], and validated the expression on neuroblastoma cell lines spiked into MNCs (Appendix A, Figure 1e). Subsequently, the staining sequence for 19 valid antibodies was determined in MELC assays followed by nuclear staining, finally resulting in a specific and robust 20-plex panel (Figure 1f, Table 1).

In conclusion, we here provide a validated 20-plex MELC panel for neuroblastoma composed of DTC markers, including a novel candidate marker called FAIM2, as well as myeloid, lymphoid, mesenchymal, and hematopoietic stem and progenitor cell markers (Figure 1f).

### 3.2. DeepFLEX Extracts Morphological Features and Protein Localization That Contribute to Cell Classification

To allow unsupervised single-cell analysis of MELC imaging data, we next developed DeepFLEX (Figure 2a), a semi-automated, deep learning-based pipeline. DeepFLEX addresses common challenges of IF-based multiplex imaging methods, such as inhomogeneous illumination, background noise, autofluorescence and unspecific binding, which are either not addressed or not effectively solved in available single-cell image analysis pipelines [76,77,78,79,80,81].

In addition to image processing, the pipeline integrates methods for deep learning-based segmentation as well as extraction, normalization and analysis of single-cell data that were recently published by our group and others (Appendix A). Single-cell data extracted by DeepFLEX describe the morphology of cells (size, perimeter) and nuclei (size, perimeter, solidity, roundness), and the protein expression in three cell compartments (cell, nucleus, cytoplasm/membrane) by three different intensity features (mean of the 20% highest pixel intensities (M20), mean intensity (ME), total intensity (TO) each (Figure 2a).

Having developed a comprehensive single-cell image analysis pipeline, we next performed an extensive feature validation and assessed the contribution of morphological features and protein marker localization to cell classification (Figure 2b–d). To this end, we performed 20-plex MELC analysis of one representative bone marrow sample with high DTC infiltration (BM 1.1, Appendix A) and applied DeepFLEX to one field of view. Next, we subsampled features and evaluated cluster stability and specificity based on a consensus clustering approach [44], reasoning that the more stable and specific the clusters are, the better suited the respective feature subset is to describe their identity. We found that, out of the three intensity features, M20 was the feature that best represented marker intensity with a mean clustering score (MCS) of 0.501. However, using all three intensity features combined gave the quantitatively (MCS = 0.560; Appendix A) and qualitatively (e.g., CD14^+^ cells in segregated cluster; Appendix A) best clustering result.

Moreover, clustering results deteriorated when morphological features were removed leaving only marker intensity to describe single cells (Appendix A), which highlights the importance of cellular and nuclear morphological features. This was supported by a principal component analysis on all cells (Figure 2c) and on DTCs only (Appendix A) showing a high contribution of morphological features to cell diversity and DTC heterogeneity, respectively.

Finally, we asked whether protein localization differs between cell types and might contribute to cell type classification using an F-test (Figure 2b) to find significantly different features (FDR-adjusted *p* < 0.05) between clusters. We observed that for CD24, the scaled mean value was higher in the cell and nucleus than in the membrane in the CD29^+^CD56^−^ cluster 3, while the opposite was true for the CD29^−^CD56^+^ cluster 4 (Figure 2d) showing that protein localization can depend on the cell type.

In conclusion, we demonstrate the relevance of morphological features for cell classification, show that protein localization can be associated with the cell type and that the best clustering result is obtained when using all three intensity features combined. We therefore considered all morphological and intensity features as well as protein localization in further analysis.

### 3.3. Single-Cell Map of Tumor Cells and the Microenvironment in Neuroblastoma Bone Marrow Metastases

To obtain a single-cell map of DTCs and the bone marrow microenvironment in patients with neuroblastoma, we performed MELC and DeepFLEX analysis on a diverse sample set of eight bone marrow samples (Appendix A) collected from three-stage M and one-stage Ms neuroblastoma patient at different time-points during therapy expected to contain a range of zero and very few to high numbers of DTCs. This resulted in an atlas of 35,700 single cells distributed between ten clusters (Figure 3a).

After confirming that potential batch effects had been eliminated (Figure 3b), we manually annotated clusters based on median feature values of cell type-specific marker proteins (Figure 3c, Appendix A) and guided by a recently published single-cell atlas [74] of healthy adult human bone marrow. In addition, we verified our annotation using representative gallery images of each cell type (Figure 3d).

We found most of the expected immune cell types including T helper cells (Th cells), cytotoxic T-lymphocytes (CTLs), monocytes and macrophages (MO/MΦ) as well as B-cells. A dominant proportion of cells in the bone marrow microenvironment of patients represented a hematopoietic mixed (Figure 3a, yellow cluster) and a stem and progenitor (Figure 3a, grey cluster) cell phenotype. Moreover, we were able to identify a segregated tumor cell cluster with co-expression of GD2, CD56, B7-H3, CD24 and FAIM2 (Figure 3a, orange cluster, and Figure 3e). The mesenchymal marker VIM showed the highest expression on monocytes and macrophages (Figure 3c and Appendix A). In bulk transcriptomic and proteomic data (Figure 1c), VIM was also highly expressed in neuroblastoma cells, which was in accordance with the IF staining results on neuroblastoma cell lines (Appendix A). However, in the eight bone marrow samples analyzed, which were prepared with the same protocol (Appendix A) and stained with the identical antibody (Table 1), DTCs were negative for VIM (Figure 3c and Appendix A), and also the mesenchymal marker CD29. This indicates that DTCs in our sample set are predominantly of an adrenergic type, but clarification will require further robust mesenchymal markers.

CD29 was enriched in two clusters (Figure 3c). The first cluster, which we annotated as myelocytes, was composed of cells with a banded nucleus, and hence low values for roundness and solidity, and a strong expression of CD24, which was shown to be expressed in immature neutrophils [82].

The second cluster (CD29^+^ cells) was negative for CD45 and hematopoietic lineage markers and mainly composed of large cells, suggesting a mesenchymal, stromal phenotype. These cells, however, displayed a low abundance of VIM. One cluster exhibited a pronounced expression of HLA-ABC, but could not be assigned to a specific cell type.

Taken together, we here provide a comprehensive representation of DTCs and the bone marrow microenvironment of neuroblastoma patients.

### 3.4. Heterogeneity of Disseminated Tumor Cells and FAIM2 as a Novel Complementary Marker

To assess tumor heterogeneity of bone marrow metastatic cells, we next performed hierarchical clustering on the single-cell data of the DTC cluster using only the DTC markers from our 20-plex MELC panel. We obtained a clustermap with 30 DTC sub-clusters ranging from clusters with very high co-expression of GD2, CD56, CD24 and CD276 to such characterized by only one or two of these markers (Figure 4a). Heterogeneous marker expression was also reflected by representative gallery images of individual DTC phenotypes (Figure 4b). Notably, half of all DTCs belonged to sub-cluster 19, which represented a dominant phenotype in the two highly tumor-infiltrated bone marrow samples BM 1.1 and BM 3.1 (63% and 2.9%, respectively; Figure 4c). While DTCs showed a round nuclear shape, cellular and nuclear size contributed to the fractionation of DTCs into distinct sub-clusters and varied between different phenotypes, e.g., 17 and 30, which was composed of large and small cells, respectively. Sub-cluster 18 and 20 displayed a high expression of FAIM2, an inhibitory protein in the Fas-apoptotic pathway of tumor cells [83,84], which was proposed as a tumor marker in small cell lung [85] and breast cancer [86]. FAIM2 is known to be primarily expressed in neurons [83,87], which was in accordance with our bulk proteomics data (Appendix A). In our RNA-seq datasets, FAIM2 transcription was significantly higher in tumor cells than in bone marrow-derived MNCs (Appendix A). Moreover, FAIM2 transcription was enriched in primary tumor cells without *MYCN* amplification as compared to those with *MYCN* amplification (Appendix A), thus supporting previous findings [88]. However, we did not observe a differential expression between these two classes in DTCs. Interestingly, in our single-cell analysis of eight neuroblastoma bone marrow samples, FAIM2 was expressed only in a subset of DTCs (Figure 3e and Figure 4a,b). As other markers that were found to be expressed by DTCs, FAIM2 was not exclusive to neuroblastoma cells, but was also found on other cell types in the bone marrow (Figure 3e and Appendix A). To assess the correlation of FAIM2 and other DTC markers, we plotted the respective marker abundances for all cells of all clusters (Figure 3e) and the DTC cluster only (Appendix A). This corroborated the observation, that only a subset of DTCs exhibit a high expression of FAIM2 along with low or intermediate abundances of the other DTC markers, while the latter, GD2, CD56, CD274 and CD24, were mostly co-expressed by DTCs. We further explored whether expression of these markers was dependent on the genetic subtype and evaluated our RNA-seq dataset for DTC marker expression in neuroblastoma with and without *MYCN*-amplification, loss of chromosome 11q or other neuroblastoma-specific genetic aberrations (Figure 4d and Appendix A). While FAIM2 and GD2 expression was constant, CD56 and CD276 were significantly lower in tumors and DTCs that carried a chromosome 11q loss, but showed no significant differential expression in other genetic subtypes, suggesting heterogeneity between patient groups with specific genetic alterations in addition to heterogeneity at the single-cell protein level.

Here we present a first exploratory survey of DTCs in bone marrow metastases on a single-cell level highlighting a hitherto unappreciated diversity pointing towards multiple distinct subclasses of DTCs. We show that FAIM2 marks a subset of DTCs and can serve as a complementary biomarker for capturing DTC heterogeneity in neuroblastoma.

### 3.5. Analysis of Bone Marrow Microenvironmental Changes

We then investigated changes in the metastatic microenvironment upon dissemination of tumor cells into the bone marrow. We first separately analyzed single-cell imaging data of bone marrow samples with no (or very low) tumor cell infiltration and compared those to bone marrow samples with high tumor cell content (Figure 5a,b and Appendix A).

Most prominently, a cluster of myelocytes appeared exclusively when DTCs were present, but not in their absence. In addition, the proportion of hematopoietic mixed as well as stem and progenitor cells, monocytes and macrophages, and undefined HLA-ABC^+^ cells was strongly reduced in samples with a high tumor cell infiltration. In contrast, we observed a relative increase in the abundance of B-cells, Th cells and CTLs (Figure 5a). In order to exclude that the myelocytic cluster, co-expressing CD24, which is also highly expressed in DTCs, and the mesenchymal marker CD29, might contain mesenchymal-type neuroblastoma cells [19,20], we performed interphase fluorescence in situ hybridization (iFISH) subsequent to MELC. The bone marrow sample with the highest DTC fraction and most abundant CD29^+^CD24^+^ cluster (BM 1.1) originated from a patient with a chromosome arm 17q gain and was therefore interrogated using a 17q-specific probe. The result (Figure 5c) unequivocally demonstrated that cells from the myelocytic cluster do not carry supernumerary 17q signals and were therefore considered normal cells that only appeared in the presence of DTCs in the bone marrow in our sample set. In addition, FISH analysis also confirmed the accurate classification of DTCs. We clearly detected six copies of 17q, which was in accordance with a previous FISH analysis of lymph node metastases from the same patient (NB1, Appendix A). As substantial differences in the bone marrow microenvironment upon tumor cell dissemination were identified by multiplex imaging, we next investigated affected cell types and functional changes in more detail. We depleted DTCs from 17 bone marrow aspirates collected at diagnosis (DTC content 1 to 70%) by magnetic activated cell sorting. Transcriptome profiles of these 17 bone marrow MNCs, i.e., the microenvironmental cell fraction, was then compared to 21 bone marrows without tumor cell infiltration (13 samples were available from Rifatbegovic et al.). Accounting for the technical limitations posed by inequalities in sample preparation, we identified 587 significantly (BH-adjusted p < 0.05 and log2FC > 1) differentially expressed genes, 353 were up-regulated and 234 down-regulated in the non-tumor fraction of tumor cell infiltrated bone marrow samples. Gene set enrichment analysis revealed a pronounced enrichment of chemotaxis and migration signatures of neutrophils and leukocytes, a pronounced inflammatory response via IFNα and γ as well as TNFα and interaction of cytokines with their receptors (Figure 5d).

In line with this, among the top 50 up-regulated genes, surface bound and secreted mediators of inflammation, e.g., IL1A and IL1B, monocyte and neutrophil chemotaxis, e.g., CXCL2 and CXCL3, but also genes indicative of immune-suppressive reactions, i.e., ARG2 and CD274 (PD-L1) were present (Figure 5e). We further interrogated cell type composition and found hematopoietic stem, myeloid and lymphoid progenitor populations decreased, whereas CD8^+^ T-cells, dendritic cell progenitors and immature neutrophils were enriched in tumor-infiltrated bone marrows (Figure 5f).

In summary, we demonstrate in two independent datasets, by a combined MELC plus DeepFLEX-based single-cell imaging and by a transcriptomic approach, that tumor cell metastasis in the bone marrow is associated with alterations in the microenvironment, specifically chemotactic cytokines and inflammatory response along with changes in the cellular composition. Most prominently, a population of immature myeloid/neutrophil cells was exclusively present in tumor-infiltrated bone marrow in our sample set.

## 4. Discussion

Herein, we provide first insights into the single-cell landscape of human bone marrow metastases including variations among DTCs as well as cells of the mesenchymal and hematopoietic compartment. The bone marrow, as part of the immune system, constitutes a niche comprised of multiple immune cell subpopulations [75], shown to be involved in cancer progression [89]. As a key regulator of hematopoietic and mesenchymal stem cell function, the niche may facilitate quiescence and drug-resistance [90], impairing current therapeutic approaches. Single-cell multi-modal analysis of healthy human bone marrow recently identified the major bone marrow mononuclear populations [74]. However, the single-cell atlas of malignant human bone marrow has so far only been described in leukemia [91,92], where the bone marrow is not considered a metastatic, but rather an originating site.

Within the bone marrow microenvironment, we observed alterations in the hematopoietic and mesenchymal cell compartment, which was dependent on tumor cell infiltration indicating that DTCs shape the metastatic niche. In support of this notion, leukemia cells are likewise known to reprogram the bone marrow niche in order to instigate changes that promote their progression [93]. Interestingly, we identified considerably fewer progenitor and other immature hematopoietic cells by multiplex imaging and transcriptomic profiling, in highly tumor infiltrated samples, which solidifies previous findings, suggesting that tumor invasion reduces the support for primitive hematopoietic stem and progenitor cells in the metastatic niche [94]. Furthermore, we found myelocytes/immature neutrophils based on protein marker expression, morphological features and transcriptome signatures only in samples with DTC infiltration. It is tempting to speculate that these may constitute a granulocytic myeloid-derived suppressor cell (G-MDSCs) type [95] or more specifically a tumor-associated neutrophil cell type, which is known to have tumor-promoting functions in several types of cancers [96]. In support of this notion, we observed increased neutrophil chemotaxis and migration signatures including upregulation of genes encoding CXCR2 ligands, which were shown to recruit neutrophils for immunosuppression and hence support of disseminated tumors cells in pre-metastatic niches [96,97,98]. It is not clear however, whether these cells might be recruited and reprogrammed from peripheral myeloid cells or whether they might be derived from bone marrow intrinsic progenitors. 

Among DTCs, we showed a high level of diversity reflected by heterogeneous cell morphologies as well as protein expression profiles and fractionation into phenotypically diverse DTC sub-clusters. Notably, half of the cells belonged to one major DTC sub-cluster, which represented a dominant phenotype in both of the two highly tumor-infiltrated bone marrow samples. This phenotype dominance was also observed in a previous study [99] on breast cancer, where in almost half of the analyzed cohort, 50% of all tumor cells belonged to a single tumor cluster, and might reflect clonal expansion, intrinsic plasticity or result from tumor-microenvironment interaction.

A subset of DTCs exhibited a high expression of FAIM2, an inhibitory protein in the Fas-apoptotic pathway, which we included into the 20-plex panel upon data mining of a previously generated neuroblastoma RNA-seq and proteomics dataset. FAIM2 was described as a therapeutic target in small cell lung cancer [85] and as a predictive marker of poor outcome in breast cancer patients [86]. Herein, we propose FAIM2 as a complementary marker to depict a broader spectrum of DTC heterogeneity, as it marked a subpopulation of DTCs and showed a lower correlation with the other analyzed DTC markers than the latter with each other. A deeper investigation of DTC sub-classes in larger patient cohorts may yield targets for neuroblastoma therapy. As neuroblastoma displays substantial genetic heterogeneity, most prominently heterogeneous amplification of the oncogenic driver *MYCN* [5,100], and since *FAIM2* expression was associated with *MYCN*-amplification status in tumors, but not DTCs, it will be important to evaluate the correlation of genetic and phenotypic heterogeneity. Especially as in response to chemotherapy or molecules inducing neuronal differentiation, such as retinoic acid or neurotrophins, tumor cells can overcome *MYCN*-driven differentiation blocks, undergo apoptosis or cellular senescence [37,101,102,103,104]. Moreover, our findings represent a valuable source of information for the design of therapeutic approaches depending on the distribution of target molecules on cancer cells, such as immunotherapies [105], and can hence contribute towards better patient stratification in neuroblastoma. 

Neuroblastoma heterogeneity has been investigated before by scRNA-seq of primary tumor samples, which were mostly composed of sympathoblast-like phenotypes and to a lesser extent of chromaffin-like and bridge cells, accounting for the embryonal trunk neural crest derived origin of neuroblastoma. However, there is a controversy about whether mesenchymal- or Schwann cell precursor like neuroblastoma cells exist and if so, whether those are more resistant to chemotherapy in vivo [19,20,25,26,27,28,29]. We previously showed that mesenchymal characteristics can be adopted upon therapy-induced tumor cell senescence [102,106]. In the present study, DTCs collected at different time points in the disease course mainly expressed markers of the sympathoblast lineage, such as GD2 [28], but did not show a mesenchymal phenotype based on the expression of the mesenchymal markers, CD29 and Vimentin. This might be explained by the limited sample as well as panel size or the fact that mesenchymal type neuroblastoma cell identity has previously been defined by master transcription factors active in gene regulatory networks. Thus, future research will require the identification of robust imaging-based markers to reliably assign neuroblastoma cells to these classes. 

Our findings were based on a multi-omics approach including single-cell analyses of MELC multiplex imaging data, enabled by the pipeline DeepFLEX, which we developed based on the integration of methods for image processing [39,40], deep learning-based segmentation [41,42], normalization [43] and single-cell analysis [46]. Current multiplex cytometry techniques comprise flow cytometry, imaging based cytometry, such as multi-epitope ligand cartography (MELC) and imaging mass cytometry (IMC), and hybrid technologies. While flow cytometry methods offer a bigger dynamic range of signal detection, morphological features are limited to cellular size and an indirect measure for granularity. Moreover, signal localization to sub-cellular compartments cannot be determined. We show that our imaging-based approach allows the analysis of cell and nuclear morphology-related features (size, roundness, solidity, perimeter) as well as the exact localization of biomarkers to e.g., the nucleus, cytoplasm and membrane, and demonstrate the relevance of those parameters for cell identity. Our approach also allows re-probing of cells for other biomarkers, e.g., genetic alterations by iFISH, which is particularly important for the unequivocal detection of tumor cells. 

DeepFLEX uses deep convolutional neural networks for nuclear and cell segmentation, which has been proven effective even for highly complex samples, showing nuclei of diverse morphologies and sizes, tight cell aggregations and overlapping cells [41,42]. DeepFLEX also tackles confounding factors of targeted imaging technologies such as unspecific binding and autofluorescence and combines deep-learning based cell and nucleus segmentation, which allow accurate single-cell assessment. The demonstrated application of DeepFLEX on MELC imaging data serves as a blueprint for further single-cell analyses by multiplex imaging methods beyond MELC, facing similar challenges. 

In our study we have chosen to investigate bone marrow aspirates. In contrast to trephine biopsies, bone marrow aspirates are more easily accessible, as collection is less invasive, cells are native, RNA is of high quality and proteins readily accessible by antibody staining without the need of harsh chemical treatment. A limitation of bone marrow aspirates is the loss of spatial context and potential bias in the cell composition as compared to the bone biopsy, since firmly attached cell types might not be released during aspiration. However, it has been previously demonstrated that bone marrow aspirates are suitable to retrieve and detect tumor cells for highly sensitive minimal residual disease detection and to profile molecular changes in tumor and non-tumor cells [4,18,34].

In conclusion, this study offers a first view into the single-cell landscape of human bone marrow metastases, which is substantially remodeled upon tumor cell invasion. This might motivate further investigations in other solid cancers with bone marrow involvement and warrant the investigation of spatio-temporal changes in tumor cell heterogeneity and microenvironment and their clinical relevance.

## Figures and Tables

**Figure 1 cancers-13-04311-f001:**
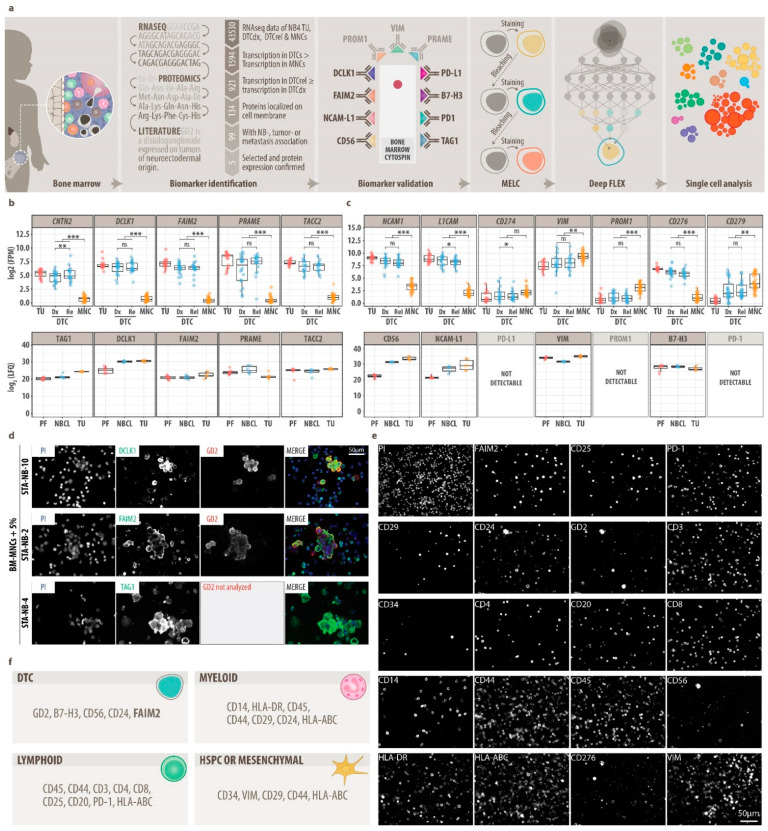
Data mining and establishment of a multiplex imaging panel (MELC). (**a**), Flow chart of experimental approach. (**b**), five potential disseminated tumor cell (DTC) biomarkers identified by data mining of RNA-seq data, proteomics data (LC-MS/MS) and literature. Top: mRNA transcription (RNA-seq) in neuroblastoma primary tumors (TU), diagnostic (dx) and relapse (rel) DTCs and bone marrow-derived mononuclear cells (BM-MNCs). DESeq2, FDR-adjusted *p* value: ns, *p* > 0.05, *, *p* ≤ 0.05; **, *p* ≤ 0.01; ***, *p* ≤ 0.001. Bottom: Protein expression in NB peripheral-nerve-associated fibroblasts (PF), neuroblastoma cells lines and TU samples. LFQ, Label-Free Quantification; FPM, fragments per million. (**c**), Extension of potential DTC biomarkers by immune checkpoint molecules (PD-L1, PD-1, B7-H3), mesenchymal-type neuroblastoma cell markers (VIM, PROM1), therapeutic target NCAM-L1 and diagnostic neuroblastoma marker CD56. DESeq2, FDR-adjusted *p* value: ns, *p* > 0.05, *, *p* ≤ 0.05; **, *p* ≤ 0.01; ***, *p* ≤ 0.001. (**d**), Representative MELC images of newly identified DTC biomarkers DCLK1, FAIM2 and TAG1 on separate samples stained by MELC. Top: DCLK1 (green) and GD2 (red) on BM-MNCs and neuroblastoma cells line STA-NB-10 (mixed 20:1); center: FAIM2 (green) and GD2 (red) on BM-MNCs and neuroblastoma cell line STA-NB-2 (20:1), bottom: TAG1 (green) on peripheral blood-derived MNCs and neuroblastoma cell line STA-NB-4 (20:1) stimulated with IFNγ and anti-CD3/28 beads. Nuclei were counterstained with DAPI (blue). (**e**), Representative MELC images of our single-cell 20-plex panel on one patient bone marrow sample. (**f**), Single-cell 20-plex panel composed of DTC, myeloid, lymphoid, mesenchymal and HSPC (hematopoietic stem and progenitor cell) markers.

**Figure 2 cancers-13-04311-f002:**
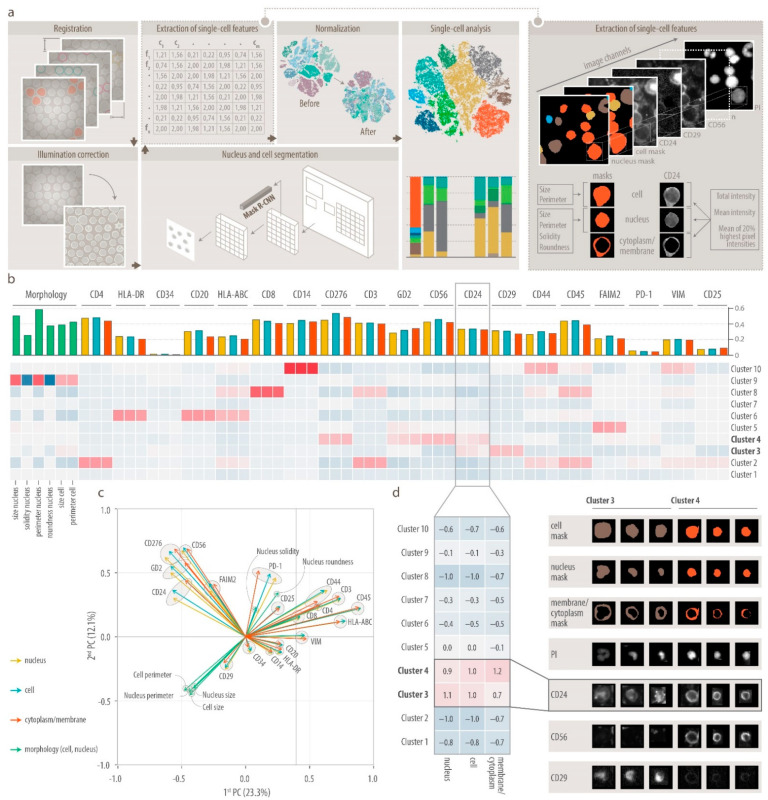
DeepFLEX and feature validation. (**a**), Schematic overview of the deep learning-based image processing (registration, illumination correction), segmentation, feature extraction, normalization and single-cell analysis pipeline DeepFLEX. Single cells are described by the intensity and morphological features. Morphological features: size and perimeter (of cell and nucleus), and solidity and roundness (of nucleus). Intensity features: total intensity, mean intensity and mean of 20% highest pixel intensities per cell compartment (cell, nucleus and cytoplasm with membrane) and marker. In total, nine intensity features per marker plus six morphological features. (**b**), Evaluating the contribution of cell compartment and morphological features to cell clustering (consensus clustering using Bioconductor package cola) using 2021 single-cell vectors, comprised of the mean of the 20% highest pixel intensities (M20) to measure marker intensity and morphological features, from one representative bone marrow (BM) sample (1.FoV of BM 1.1, Appendix A). Signature heatmap showing scaled mean per cluster and feature. Barchart showing the maximal difference between significantly (F-test with FDR-corrected *p*-value < 0.05) different clusters per feature. Green, morphological features; blue, M20 in cell: yellow; M20 in nucleus; red, M20 in cytoplasm with membrane. (**c**), Biplot representation of principal component analysis (PCA) on the autoscaled data (single-cell vectors, comprised of M20 of each marker and morphological features) in one representative BM sample (1.FoV of BM 1.1), showing the projection of the data set in the PC1xPC2 plane. Length of adjacent and opposite leg of arrows show contribution to first and second principal components, respectively. Green, morphological features; blue, M20 in cell: yellow; M20 in nucleus; red, M20 in cytoplasm/membrane. (**d**), Left, scaled mean of CD24 marker intensity (M20) higher in cell and nucleus than in cytoplasm/membrane in cluster 3 and vice versa in cluster 4; right, Gallery images of cell, nucleus and cytoplasm/membrane segmentation masks, and propidium iodide (PI), CD24, CD56 and CD29 MELC stainings for three representative cells from cluster 3 and 4.

**Figure 3 cancers-13-04311-f003:**
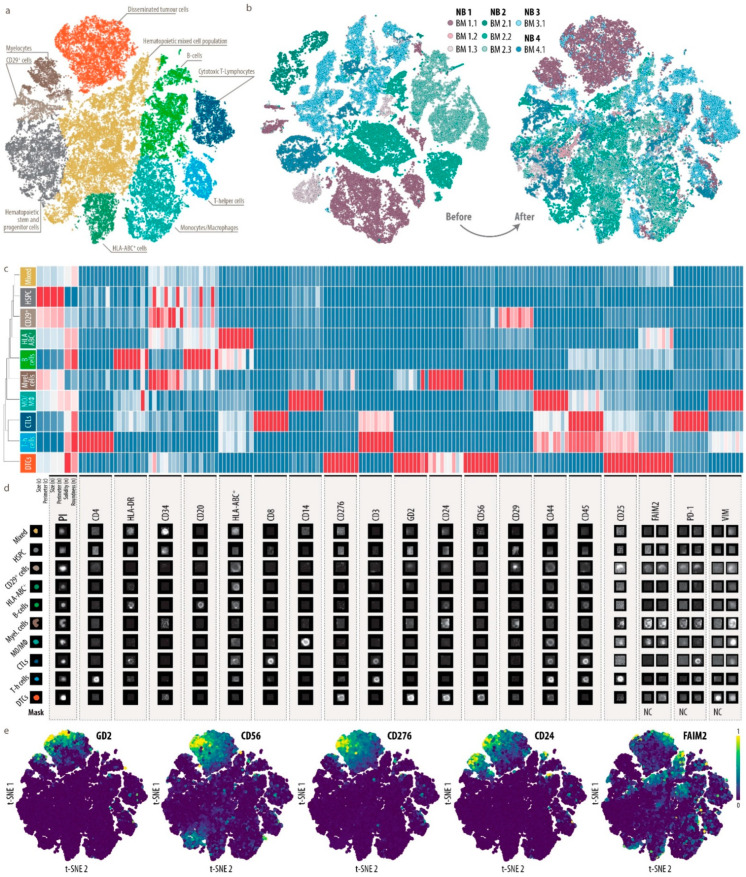
Single-cell map of eight bone marrow metastases from four patients with stage M and Ms neuroblastoma. (**a**), Single-cell atlas of 35,700 single cells clustered colored by cell type. Dimensionality reduction was performed by A-tSNE (approximated and user steerable t-distributed Stochastic Neighbor Embedding) and subsequent clustering by GMS (Gaussian Mean Shift) in Cytosplore. (**b**), A-tSNE plot of 35,700 single cells colored by sample before and after batch correction using RESTORE [43]. (**c**), Heatmap showing the median feature values of all created clusters with feature-wise scaling. *n*, nucleus; c, cell. Nine columns per marker represent, from left to right, mean intensity, total intensity and mean of the highest 20% of pixel values in the (I) nucleus, (II) cell and (III) cytoplasm/membrane. DTCs, disseminated tumor cells; Myel., myelocytes; MO/MΦ, monocytes/macrophages; HSPC, hematopoietic stem and progenitor cells; T-h cells, T-helper cells; CTLs; cytotoxic T-lymphocytes; Mixed, hematopoietic mixed cell population. (**d**), Representative gallery images of all cell types. For FAIM2, PD-1 and VIM we introduced negative controls (NC) to be used for normalization during data processing. Hence, for these four biomarkers, the ratio between right column and left column (NC) represents the true signal. (**e**), A-tSNE plot of 35,700 single cells colored by GD2, CD56, CD276, CD24 and FAIM2 signal intensity (mean of the highest 20% of pixel values in the cytoplasm/membrane).

**Figure 4 cancers-13-04311-f004:**
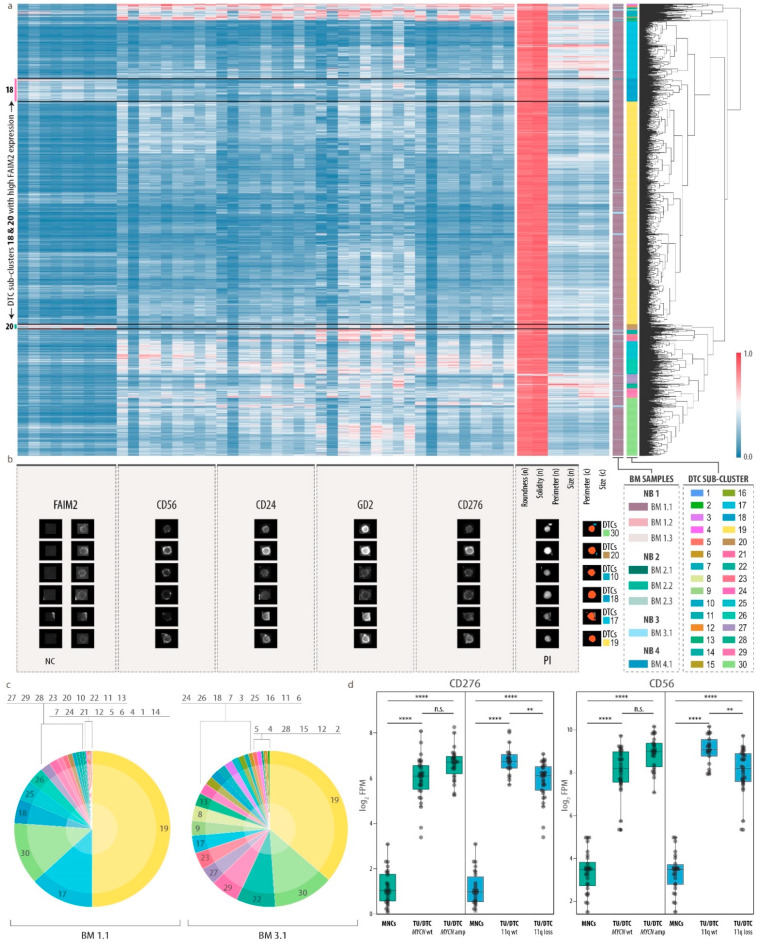
Characterization of DTC heterogeneity and qualification of FAIM2 as a novel complementary biomarker. (**a**), Cluster map (hierarchical clustering by Voorhees) showing normalized single-cell feature values of DTCs. *n*, nucleus; c, cell. Nine columns per marker represent, from right to left, mean intensity, total intensity and mean of the highest 20% of pixel values in the (I) nucleus, (II) cell and (III) cytoplasm/membrane. Color bar on the right shows 30 sub-clusters. Color bar on the left shows the corresponding bone marrow sample. (**b**), Representative gallery images of six selected cells from different DTC sub-clusters reflecting DTC heterogeneity. For FAIM2 we introduced negative controls (NC) to be used as background threshold levels during data processing. Hence, for this biomarker, the ratio between right column and left column (NC) represents the true signal. (**c**), Proportion of 30 DTC sub-clusters in highly tumor-infiltrated bone marrow samples (BM 1.1, BM 3.1). (**d**), CD56 and CD276 mRNA transcription in bone marrow-derived mononuclear cells (MNCs) and neuroblastoma tumor cells (TU/DTC) without (wt) and with (amp/loss) genetic aberration. Wilcoxon–Mann–Whitney with FDR-corrected *p*-values: ns, **, *p* ≤ 0.01; ****, *p* ≤ 0.0001. FPM, fragments per million; amp, amplification.

**Figure 5 cancers-13-04311-f005:**
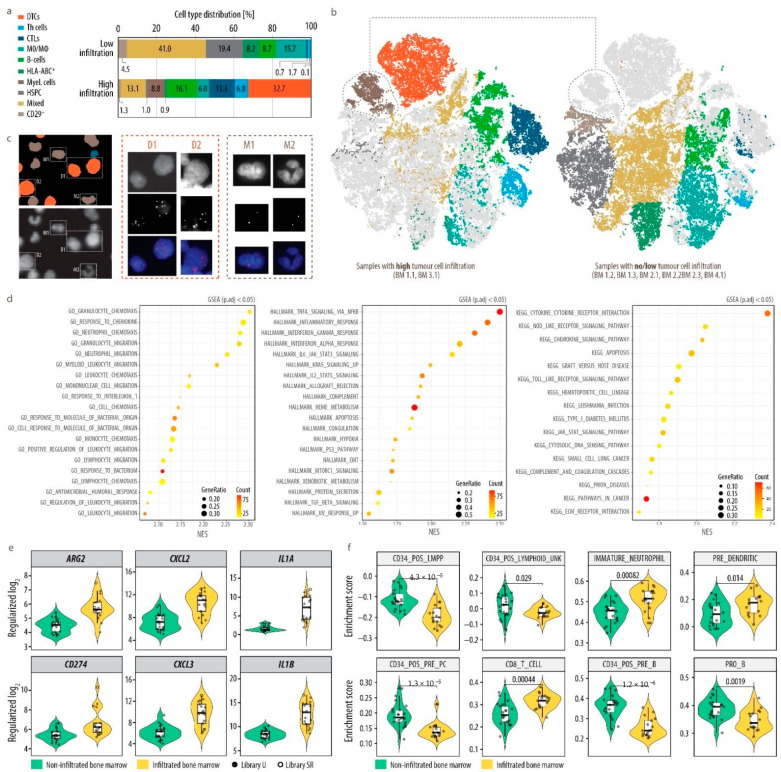
Changes in the cell composition associated with the presence of tumor cells in the bone marrow. (**a**), Bar charts demonstrating the cell composition in bone marrow samples with no/low and high tumor cell infiltration. DTCs, disseminated tumor cells; Myel., myelocytes; MO/MΦ, monocytes/macrophages; Mes. cells, mesenchymal cells; HSPC, hematopoietic stem and progenitor cells; T-h cells, T-helper cells; CTLs; cytotoxic T-lymphocytes; Mixed, hematopoietic mixed cell population. (**b**), A-tSNE plot of 35,700 single cells highlighted by samples with high (left, BM 1.1, BM 3.1) and no/low (right, BM 1.2, BM 1.3, BM 2.1, BM 2.2, BM 2.3, BM 4.1) tumor cell infiltration and colored by cell type. Dimensionality reduction was performed by A-tSNE (approximated and user steerable t-distributed Stochastic Neighbor Embedding) and subsequent clustering by GMS (Gaussian Mean Shift) in Cytosplore. (**c**), FISH analysis with chromosome 17q-specific probe on MELC-preprocessed sample BM 1.1, collected from a patient with 17q gain. Nucleus segmentation mask (left, top) pseudo-colored according to cell type and based on propidium iodide image (right, bottom) acquired during MELC. Six copies of 17q (red) were detected on DTCs (D, orange) and 2 on myelocytes (M, brown). The 17p reference probe did not yield interpretable results due to preprocessing of the sample by MELC. Nuclei were counterstained with DAPI (blue). (**d**), RNA-seq analysis of MACS separated GD2 negative bone marrow MNC fraction of DTC-infiltrated (*n* = 17) and non-infiltrated (*n* = 21) bone marrow. Top 20 significantly (p.adj < 0.05; p.adj = BH-adjusted *p*-value) positively enriched (NES > 0; NES = normalized enrichment score) gene sets from GOBPs (GO-biological processes), hallmark and KEGG gene set collections as determined by GSEA (gene set enrichment analysis) (Appendix A) performed on log2FC ranked gene list are shown for bulk RNAseq data of the MNC fraction from tumor-infiltrated vs. non-infiltrated bone marrow. Count—core_enrichment size; GeneRatio—core_enrichment size divided by setSize. (**e**), six genes implicated in inflammation, immuno-suppression and neutrophil attraction from the top 50 significantly (p.adj < 0.05 and log2FC > 1; p.adj = BH-adjusted *p*-value) upregulated genes (Appendix A) in bulk RNAseq data of the MNC fraction from tumor-infiltrated vs. non-infiltrated bone marrow. Regularized log2, regularized logarithm (rlog) transformed counts normalized with respect to library size; U, unstranded; SR, stranded protocol. (**f**), Single-sample gene set enrichment analysis (ssGSEA) of selected gene sets for bone marrow-derived cell types (MSigDB—C8: HAY_BONE_MARROW_) in bulk RNAseq data of the MNC fraction from tumor-infiltrated vs. non-infiltrated bone marrow. Enrichment score—normalized enrichment score as calculated by GSVA ssgsea algorithm, *p*-values—calculated using unpaired two-samples Wilcoxon test (R function wilcox.test()).

**Table 1 cancers-13-04311-t001:** Validated 20-plex antibody panel and MELC imaging sequence. All primary and secondary antibodies, which passed the validation procedure and were included in the final 20-plex MELC panel. Negative control secondary antibodies were implemented, which were applied to the sample prior to indirect staining of the respective primary antibody. n.r., not relevant.

Step	Antibody	Conjugate	Class|Host|Isotype	Clone	Supplier	Catalogue-Number	Optimal Dilution
	Sw α Rb	FITC	polyclonal swine IgG	polyclonal	Dako	F0205	1:50
1	FAIM2	unconj.	polyclonal rabbit IgG	polyclonal	ThermoFisher	PA5-20311	1:50
	Sw α Rb	FITC	polyclonal swine IgG	polyclonal	Dako	F0205	1:50
2	CD25	PE	monoclonal mouse IgG	HI25a	ImmunoTools	21810254	1:20
	Ms α Biot.	Cy3	monoclonal mouse IgG	3D6.6	Jackson ImmunoResearch	200-162-211	1:800
3	PD-1	biotinylated	monoclonal mouse IgG1	NAT105	BioLegend	367418	1:50
	Ms α Biot.	Cy3	monoclonal mouse IgG	3D6.6	Jackson ImmunoResearch	200-162-211	1:800
4	CD29	FITC	monoclonal mouse IgG1	HI29a	ImmunoTools	21810293	1:20
5	CD24	FITC	monoclonal mouse IgG1	SN3	ImmunoTools	21270243	1:20
6	GD2	FITC	monoclonal chinese hamster/humanized	ch14.18/deltaCHO	Tübingen	n.r.	1:100
7	CD3	PE	monoclonal mouse IgG1	UCHT1	ImmunoTools	21620034	1:20
8	CD34	PE	monoclonal mouse IgG1	4H11[APG]	ImmunoTools	21270344	1:20
9	CD4	PE	monoclonal mouse IgG2a,k	VIT4	Miltenyi Biotec	130-113-214	1:20
10	CD20	PE	recombinant human IgG1	REA780	Miltenyi Biotec	130-111-338	1:20
11	CD8	PE	monoclonal mouse IgG1	HIT8a	ImmunoTools	21810084	1:20
12	CD14	PE	monoclonal mouse IgG1	18D11	ImmunoTools	21620144	1:20
13	CD44	PE	monoclonal rat IgG2b	IM7	ImmunoTools	21850444	1:20
14	CD45	PE	monoclonal mouse IgG1	HI30	ImmunoTools	21810454	1:20
15	CD56	PE	monoclonal mouse IgG1	B-A19	ImmunoTools	21810564S	1:20
16	HLA-DR	PE	monoclonal mouse IgG1	HI43	ImmunoTools	21819984	1:20
17	HLA-ABC	PE	monoclonal mouse IgG2a	W6/32	ImmunoTools	21159034	1:20
18	B7-H3	PE	human IgG1	REA1094	Miltenyi Biotec	130-118-570	1:40
	Gt α Ch	FITC	polyclonal goat IgG	polyclonal	ThermoFisher	A16055	1:500
19	Vimentin	unconj.	recombinant chicken IgY	polyclonal	Milipore/Chemicon	AB5733	1:100
	Gt α Ch	FITC	polyclonal goat IgG	polyclonal	ThermoFisher	A16055	1:500
20	Propidium Iodide	PI	n.r.	n.r.	Genaxxon bioscience	M3181.0010	1:1000

## Data Availability

The RNA-seq datasets used for data mining and RNA-sequencing analysis are available for download on the GEO data repository under accession numbers GSE94035 and GSE172184. The mass spectrometry proteomics data has been deposited to the ProteomeXchange Consortium (proteomecentral.proteomexchange.org, accessed on 21 July 2021) via the PRIDE partner repository (ebi.ac.uk/pride/, PMID: 24727771) with the dataset identifier PXD018267. Python code for the DeepFLEX pipeline is available on https://github.com/perlfloccri/DeepFLEX, accessed on 21 July 2021. A compiled release with all necessary dependencies pre-installed is available from dockerhub URL https://hub.docker.com/repository/docker/imageprocessing29092020/deepflex, accessed on 21 July 2021. The MELC multiplex imaging data of our neuroblastoma cohort is available at https://doi.org/10.5281/zenodo.5906989, accessed on accessed on 21 July 2021. The R code for the RNAseq analysis is available on https://github.com/prepiscak/neuroblastoma_BM_infiltration, accessed on 21 July 2021.

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
