# Peer review of "Landscape of Bone Marrow Metastasis in Human Neuroblastoma Unraveled by Transcriptomics and Deep Multiplex Imaging"

_cancers, 2021, doi:10.3390/cancers13174311_

Round 1

Reviewer 1 Report

The aim of the paper is to study the tumor heterogeneity and microenvironmental changes in neuroblastoma involving bone marrow. The study design is good, with exhaustive M&M section. The results are well reported and the conclusions coherent. In this view the paper is very dense and requires an adequate reading time.

The aim of the paper is to study the tumor heterogeneity and microenvironmental changes in neuroblastoma involving bone marrow.

The authors perform an original in-depth analysis of how metastatic cells shape the bone marrow microenvironment, on a single-cell level. Furthermore, they found that FAIM2 marks a subset of disseminated tumor cells, potentially serving as a complementary biomarker for capturing heterogeneity in neuroblastoma.

The study design and methodology is solid and exhaustively reported.

The conclusions are coherent with the results and address the aim of the paper.

The references are appropriate.

Reviewer 2 Report

Lazic et al identified several factors that are found in the tumour microenvironment of bone marrow metastasis. Bone marrow metastasis were studied by interrogating multi-omics datasets, including multiplex image analysis, and performing transcriptomics of 38 bone marrow aspirates. Disseminated tumour cells were associated with an inflammatory response and immuno-suppressive cell types. Neuroblastoma was used as an example as it is the most common solid childhood cancer and has a high bone metastasis component at the advanced stage, as well as high intra-tumour heterogeneity.

1. Please provide a simple/lay summary of the presented work.

2. Use instead of ‘children’ for example on page 5, line 199, the word ‘patients’ as it was used for example in the abstract, line 22.

3. The authors could comment on how their data can be used to identify treatment targets that address developmental aberrations and overcome differentiation blocks. Some of their previous findings on MYCN-amplified neuroblastoma should be considered here as well.

4. To which extent can the findings be used to design immunotherapies, or personalized treatments, tuned to the immune landscape of bone metastasis.

5. There are 5 figures and 2 tables, as well as 18 supplementary figures. Maybe some of the key results from the supplementary figures could be shown as an actual figure. Tables 1 and 2 could be moved to the supplement as they are related to the materials and methods and don’t show any results.

6. The methods description could be shortened to provide the main technologies or analyses in the text and the rest as supplementary methods. This would improve the readability for a broader audience.
